# Molecular Characterization of Community- and Hospital- Acquired Methicillin-Resistant *Staphylococcus aureus* Isolates during COVID-19 Pandemic

**DOI:** 10.3390/antibiotics12010157

**Published:** 2023-01-12

**Authors:** Muhammad Sohail, Moazza Muzzammil, Moaz Ahmad, Sabahat Rehman, Mohammed Garout, Taghreed M. Khojah, Kholoud M. Al-Eisa, Samar A. Breagesh, Rola M. Al Hamdan, Halimah I. Alibrahim, Zainab A. Alsoliabi, Ali A. Rabaan, Naveed Ahmed

**Affiliations:** 1Department of Microbiology, Chughtai Lab, Lahore 54000, Pakistan; 2Department of Medical Lab Technology, Faculty of Rehabilitation & Allied Health Sciences, Riphah International University Islamabad (QIE Campus), Lahore 54000, Pakistan; 3Department of Medical Education, King Edward Medical University, Lahore 54000, Pakistan; 4Department of Pathology, HITEC Institute of Medical Sciences, Heavy Industries Taxila Cantt, Taxila 47070, Pakistan; 5Department of Community Medicine and Health Care for Pilgrims, Faculty of Medicine, Umm Al-Qura University, Makkah 21955, Saudi Arabia; 6Molecular Biology Department, Riyadh Regional Laboratory, Riyadh 11425, Saudi Arabia; 7Pharmacy Department, Qatif Central Hospital, Qatif 32654, Saudi Arabia; 8Molecular Diagnostic Laboratory, Johns Hopkins Aramco Healthcare, Dhahran 31311, Saudi Arabia; 9College of Medicine, Alfaisal University, Riyadh 11533, Saudi Arabia; 10Department of Public Health and Nutrition, The University of Haripur, Haripur 22610, Pakistan; 11Department of Medical Microbiology and Parasitology, School of Medical Sciences, Universiti Sains Malaysia, Kubang Kerian 16150, Malaysia

**Keywords:** hospital-acquired infections, biofilm formation, adhesion genes, MRSA, antimicrobial resistance, AMR

## Abstract

Methicillin-resistant *Staphylococcus aureus* (MRSA) is a drug-resistant superbug that causes various types of community- and hospital-acquired infectious diseases. The current study was aimed to see the genetic characteristics and gene expression of MRSA isolates of nosocomial origin. A total of 221 MRSA isolates were identified from 2965 clinical samples. To identify the bacterial isolates, the clinical samples were inoculated on blood agar media plates first and incubated at 37 °C for 18–24 h. For further identification, the Gram staining and various biochemical tests were performed once the colonies appeared on the inoculated agar plates. The phenotypic identification of antibiotic susceptibility patterns was carried out using Kirby–Bauer disk diffusion method by following the Clinical and Laboratory Standards Institute (CLSI) 2019 guidelines. The biofilm-producing potentials of MRSA were checked quantitatively using a spectrophotometric assay. All strains were characterized genotypically by *SCCmec* and *agr* typing using the specific gene primers. Furthermore, a total of twelve adhesion genes were amplified in all MRSA isolates. MRSA was a frequently isolated pathogen (44% community acquired (CA)-MRSA and 56% hospital acquired (HA)-MRSA), respectively. Most of the MRSA isolates were weak biofilm producers (78%), followed by moderate (25%) and strong (7%) biofilm producers, respectively. Prominent adhesion genes were *clfB* (100%), *icaAD* (91%), *fib* (91%), *sdrC* (91%) followed by *eno* (89%), *fnbA* (77%), *sdrE* (67%), *icaBC* (65%), *clfA* (65%), *fnbB* (57%), *sdrD* (57%), and *cna* (48%), respectively. The results of the current study will help to understand and manage the spectrum of biofilm-producing MRSA-associated hospital-acquired infections and to provide potential molecular candidates for the identification of biofilm-producing MRSA.

## 1. Introduction

Methicillin-resistant *Staphylococcus aureus* (MRSA) is a notorious multidrug resistant (MDR) superbug, commonly associated with hospital-acquired infections (HAIs) [1,2], and these infections are always challenging to treat through a single intervention [3]. Previous studies have revealed that severity and diversity of infection caused by MRSA depend upon the combination of a set of adhesion genes and molecular typing of the pathogen [4,5]. It is believed that MRSA evolved gradually and acquired the antibacterial resistance inducing mobile genetic elements and biofilm production, which makes the MRSA a superbug of hospital-acquired (HA-MRSA) and community-acquired (CA-MRSA) infections [6].

Molecular typing involving SCC*mec*, *agr* typing, and microbial surface components identifying adhesive matrix molecules (MSCRAMM) are vital factors to depict the relatedness of MRSA and predict the severity of infection [7]. The most common MSCRAMM are clumping factor A and B (*clfA*, *clfB*), intercellular adhesin (*icaAD*, *icaBC*), laminin-binding protein (*eno*), fibronectin-binding proteins A and B (*fnbA*, *fnbB*), fibrinogen-binding protein (*fib*), serine–aspartate repeat proteins C, D and E (*sdrC*, *sdrD*, *sdrE*), and collagen-binding adhesin (*cna*), respectively [8]. Sequence type ST239 was the most frequently isolated HA-MRSA, whereas ST-59 was the most frequently isolated clone among CA-MRSA worldwide [9]. Similarly, clonal complex CC8, CC1, and sequence type ST8 and ST30, respectively, are most frequently reported in Pakistan [10].

Biofilm is a complex mechanism of bacterial communities that poses extraordinary resistance to antibiotics, persistent infections, and enhances the survival of pathogens inside the human body over extended periods. Biofilm formation is a multistep process that depends upon bacterial phenotypical characteristics, gene expression, and adhesion proteins present on the surface of bacteria [11,12]. Some MRSA strains having specific genotyping characteristics can form biofilm various infection sites [13]. 

The attachment, maturation, and dispersion are the three phases of biofilm development. The attachment phase may also be divided into the first reversible attachment and the irreversible attachment stages [12]. Stronger physical or chemical shear pressures may be tolerated by the irreversibly attached biofilm. The first stage of biofilm development for the human infections by *Staphylococcus aureus* (*S. aureus*) is adhesion to human matrix proteins including fibronectin (Fn), fibrinogen (Fg), and vitronectin (Vn), etc. [14]. The peptidoglycan on the cell wall is covalently linked to microbial surface components that recognized adhesive matrix molecules dependent adhesions. Cells of the same species or cells from other species are attracted to the biofilm from the bulk fluid after the initial layer of the biofilm has been formed. A thin layer of biofilm develops into a mushroom or tower-shaped structure. Bacteria are arranged in a thick biofilm (>100 layers) in accordance with their metabolism and aero tolerance [15]. The imprisoned bacteria produce additional biofilm scaffolds as the biofilm develops, including proteins, DNA, polysaccharides, etc. The dispersion phase, which is similarly crucial for the biofilm life cycle, comes after biofilm maturation. There are several factors for biofilms dispersion, including nutritional deficiency, stiff competition, population growth, etc. The whole biofilm may experience dispersal, or just a part of it. The emergence of new biofilms at different places is promoted by the release of planktonic bacteria [16,17].

While MRSA is one of the most important pathogens worldwide, some strains are restricted to one geographic area. This makes molecular characterization of great importance for investigating the epidemiology of MRSA both locally and globally. Biofilm formation is considered a virulence property of *S. aureus* and results in recurrent infections that are difficult to treat and lead to higher treatment costs. Therefore, the current study was conducted with the primary objective to see the overall prevalence of bacterial infections among possible community- and hospital-acquired infection during the COVID-19 pandemic. Simultaneously, the secondary objective was to investigate the phenotypic and genotyping characterization of MRSA isolates.

## 2. Materials and Methods

### 2.1. Study Design and Study Setting

Before starting the study, an ethical approval was obtained from the Institutional Review Board of Chughtai Lab, Lahore. The current study was conducted by the Department of Microbiology, Chughtai Lab, Lahore in collaboration with the Department of Medical Lab Technology, Faculty of Rehabilitation & Allied Health Sciences, Riphah International University Islamabad, QIE Campus Lahore, Pakistan, from the duration of 16 November 2021 to 4 March 2022. A total of 2965 clinical samples (blood, urine, sputum, tracheal aspirate, bronchoalveolar lavage, abscess, and wound swabs, respectively) were received from the different public and private sector hospitals for microbiological diagnosis. After the collection of samples, these were transported to the microbiology section of Chughtai Lab for further processing. Chughtai Lab is a private sector laboratory which has collection centers and stat labs throughout the country, providing the facilities for radiological and laboratory diagnosis.

At first, the clinical samples suspected of bacterial infection were processed in the Department of Microbiology, Chughtai Lab, for microbiological diagnosis, and then after the final diagnosis, only the MRSA isolates were collected and transported to the Department of Medical Lab Technology, Riphah International University Islamabad, QIE Campus Lahore for further molecular-based processing.

### 2.2. Data Collection

The data of patients was collected from the hospitals retrospectively using a pre-structured questionnaire. The data includes the coronavirus diseases-2019 (COVID-19) status of patients, and their comorbidities. The data about COVID-19 includes the current or previous history of acquiring COVID-19 infections and states if during the current study, the patients were tested for COVID-19 or not.

### 2.3. Pathogen Isolation and Identification 

Following the standard microbiological techniques, samples were analyzed in the microbiology laboratory to isolate and identify the bacterial pathogens. After receiving the clinical samples in the microbiology laboratory, the samples were inoculated on blood and chocolate agar media plates with the sterile disposable wire loop and incubated at 37 °C for the period of 18–24 h. After the incubation period, the inoculated plates were taken out from the incubator and observed for the presence of bacterial colonies. The bacterial identification and confirmation was carried out by bacterial colony morphology, Gram staining, and biochemical (catalase, citrate, and coagulase) tests [18]. Once the bacterial colonies were identified, the isolated colonies were proceeded for antibiotic susceptibility testing.

### 2.4. Antimicrobial Sensitivity Testing

The antibiotic susceptibility testing was performed by following the guidelines from Clinical and Laboratory Standards Institute (CLSI) 2019. The antibiotic sensitivity testing of bacterial isolates was performed on Mueller-Hinton agar (Oxoid, UK) using the Kirby–Bauer disk diffusion method [19]. The panel of antibiotics against each of the isolated bacterium was selected based on the CLSI-2019 guidelines with certain extra drugs (amikacin and tobramycin). The tested isolates were declared sensitive or resistant by following the established criteria of the zone of inhibition (ZOIs) sizes around tested disk of antibiotic. The ZOIs of each tested antibiotic were given in CLSI guideline 2019 [19]. 

The tested antibiotics were amikacin (30 µg), chloramphenicol (30 µg), cefoxitin (30 µg), ciprofloxacin (5 µg), doxycycline (30 µg), ofloxacin (5 µg), trimethoprim/Sulfamethoxazole (23.75 µg), clindamycin (2 µg), azithromycin (15 µg), gentamicin (10 µg), linezolid (30 µg), tobramycin (10 µg), and vancomycin, respectively. The susceptibility pattern of vancomycin was checked by minimum inhibitory concentration (MIC) testing.

### 2.5. Biofilm Formation

The biofilm formation potential of MRSA isolates was assayed quantitatively using a spectrophotometric assay. Positive control included *Staphylococcus epidermidis* ATCC 12228. Standard bacterial inoculum (0.5 McFarland) was prepared in Blood Head Infusion (BHI) (Oxoid, UK) supplemented with 1% glucose (Sigma-Aldrich, Burlington, USA) and inoculated in 96 well plate. This polystyrene microtiter plate was incubated for 48 h at 37 °C without agitation. The cells were washed with physiological saline (Thermo Fisher Scientific, Waltham, MA, USA) three times and stained with 0.1% CV (crystal violet) (Sigma-Aldrich, Germany). After washing with physiological saline, the stain was dissolved in 200 µL of 95% ethanol and measured at 595 nm by an ELISA plate reader (Rayto, Shenzhen, China). This assay was performed in triplicate. Based on OD^595^, biofilm formation capability was categorized as non-biofilm producers OD < 0.05, weak-biofilm producers OD > 0.5 and ≤1, moderate-biofilm producers OD > 1 and ≤2, and strong-biofilm producers having OD > 2, respectively [20]. 

### 2.6. Genotyping of MRSA

All strains were characterized genotypically for SCC*mec* [21] and *agr* typing [22] by using the primers and PCR conditions previously reported [23]. Twelve adhesion genes including *clfA*, *clfB*, *icaAD*, *icaBC*, *eno*, *fnbA*, *fnbB*, fib, *sdrC*, *sdrD*, *sdrE* and *cna* were amplified in all MRSA isolates following previously reported sets of primers and PCR conditions [10]. Sixteen strains of MRSA selected based on the antibiogram, SCC*mec*, and *agr* typing for MLST (Multilocus sequence typing) followed by gene expression studies [24]. 

### 2.7. Data Analysis

Statistical analysis was performed by employing the data in SPSS software version 22.0 (Chicago, IL, USA). The mean, standard deviation (SD) and percentages (%) were calculated.

## 3. Results

### 3.1. Isolation of MRSA

A total of 2965 clinical samples from 2692 patients for possible bacterial infection were analyzed to identify the pathogens and their antibiotic susceptibility patterns. From these 2965 samples, in total, 2637 were found to be positive for different bacterial infections. In total, 336 samples were found positive for two or three types of bacterial infections. The growth of *S. aureus* was seen in 24.82% (736/2965) of samples; among them, 30.02% (221/736) were MRSA. The prevalence of other pathogens is shown in Table 1. 

From the 2692 patients included in the current study, a total of 876 were found positive for COVID-19 infection during the study time. A total of 1789 patients were hospitalized, whereas the remaining 903 were checked clinically by the clinical/physician, given medicines, and not hospitalized. Among the 876 COVID-19 patients, a total of 241 (27.51%) were found positive for bacterial infections. The prevalence of MRSA among COVID-19 patients who were coinfected with different bacterial infections was 17.4% (*n* = 42).

### 3.2. Antimicrobial Sensitivity Testing (AST)

The antimicrobial sensitivity profile of 221 strains of MRSA was determined following the standard concentration of antibiotics recommended by CLSI. All strains of MRSA were sensitive to vancomycin and linezolid. MRSA strains were susceptible to chloramphenicol (79%) and doxycycline (59%). On the other hand, MRSA was highly resistant to amikacin (81%), gentamycin (93%), tobramycin (96%), azithromycin (97%), ciprofloxacin (96%), ofloxacin (96%), Trimethoprim/Sulfamethoxazole (89%), and clindamycin (85%). The prevalence of CA-MRSA and HA-MRSA is 44% (98/221) and 56% (123/221) based upon genotypic characteristics. HA-MRSA are more resistant to tested antibiotics, especially amikacin, doxycycline, trimethoprim/sulfamethoxazole, and clindamycin compared to CA-MRSA except for chloramphenicol (Figure 1).

### 3.3. Biofilm Assay

#### 3.3.1. Congo Red Agar

MRSA showed varying degrees of slime production from very black colonies (7%), black (15%), and light back or pink colonies (78%), respectively. ATCC 35556 *S. aureus* produced very black colonies after 48 h of incubations, used as a positive reference strain, and ATCC 1228 was used as a negative control.

#### 3.3.2. Quantitative Microtiter Plate Method

All MRSA, 100% strains, showed biofilm production with varying degree of adhesion from strong 7% (Optical density at 595 >1.0 nm), moderate 15% (Optical density at 595 < 1.0 nm and >0.6 nm), to weak adhesion 78% (Optical density at 595 < 0.6 nm). Reference strain ATCC 35556 *S. aureus* firmly adhered to the microtiter plate, whereas ATCC 1228 did not adhere to the plate. Based upon an antibiogram, most resistant strains of MRSA (*n* = 16) were selected for molecular studies. Biofilm production by CA-MRSA and HA-MRSA is illustrated in Figure 2.

### 3.4. Detection of Biofilm Genes

Twelve biofilm-associated adhesion genes were detected among 221 strains of MRSA isolates. The prevalence of 12 gene involved in biofilm production is: *clfA* (65%), *clfB* (100%), *icaAD* (91%), *icaBC* (65%), *eno* (89%), *fnbA* (77%), *fnbB* (57%), *fib* (91%), *sdrC* (91%), *sdrD* (57%), *sdrE* (67%), and *cna* (48%), respectively. 

### 3.5. Molecular Characterization of MRSA

SCC*mec* and *agr* typing elucidated that MRSA strains isolates are genetically diverse. Most of the strains were classified as SCC*mec* type II (20%), type III (17%), type IV (35%), type V (6%), and type VI (2%), and *agr* type I (40%), type II (8%), type III (5%), and type IV (4%). Some strains were not typed by SCC*mec* or/and *agr* typing. No statistically significant difference was found between biofilm formation protentional and SCC*mec* or *agr* typing. The correlation of biofilm formation protentional, adhesion genes, SCC*mec*, and *agr* typing is illustrated in Table 2. 

Genetic characterization of 16 MRSA strains was conducted to elucidate the relationship between biofilm formation potential and clonal lineages. Molecular analysis revealed that ST2490 (5/16) was the most frequent type, followed by ST8 (3/16), ST5 (2/16), and ST72 (2/16) that are responsible for biofilm formation.

## 4. Discussion

The infections by superbugs such as MRSA have always been a substantial threat to public health services and the healthcare system [25,26]. The HAIs are notoriously challenging to treat because of emerging and inherited antimicrobial resistance (AMR), biofilm formation, and low penetration at the site of infection [27]. The current study found that *S. aureus* isolates*,* primarily the MRSA, are predominantly associated with HAIs. The majority of MRSA (>50%) were resistant to available recommended antibiotics except for linezolid and vancomycin, to which MRSA was 100% sensitive. HA-MRSA was more resistant to tested antibiotics than CA-MRSA, especially amikacin, doxycycline, trimethoprim/sulfamethoxazole, and clindamycin. On the other hand, CA-MRSA was a strong biofilm producer as compared to HA-MRSA.

A previous study from the United Kingdom (UK) and Europe concluded that MRSA was a significant (>40%) pathogen for different infections [28]. A study from Lahore, Pakistan showed that the MRSA was 14.9% prevalent during the COVID-19 pandemic [29]. Some studies [30,31] also reported CONS as a significant pathogen for different infections. However, results of the current study showed the majority of the infections by *S. aureus* were caused MSSA strains (16.85%). This difference in prevalence is justifiable based on surgical practices and the tendency to review cases of surgical procedures and health facilities in general. 

A study conducted on 105 strains of *S. aureus* isolated from various clinical specimens at Kabul University, Afghanistan concluded that all MRSA were sensitive to vancomycin, and 8.5% of MRSA were resistant to clindamycin; similar investigations were reported in the current study [32]. There are studies conducted in China, United States, South Africa, Japan, and Australia which reported linezolid and vancomycin susceptivity is 100% towards MRSA isolated from clinical specimens such as pus, bone and joint infections, and prosthetic device [33,34]. A study from India reported 1% resistance to linezolid and vancomycin [15]. Another study reported that the MRSA isolated from skin and soft tissue infection were 15% susceptible to ciprofloxacin, 53% erythromycin, 77% clindamycin, 86% doxycycline, 67% gentamicin, 48% cotrimoxazole, and 94% chloramphenicol [35], respectively; the results of this study were opposite to the current results - possibly due to the nature of specimens, empirical therapy, and guidelines for the usage of antibiotics for MRSA infections. On the other hand, it also reported that CA-MRSA was more sensitive to antibiotics compared with HA-MRSA, in agreement with the current study [35]. A study published recently on the antibiogram of *S. aureus* in Pakistan reported 1% resistance to linezolid, 2% to vancomycin, 16% to chloramphenicol, 42% to doxycycline, 56 % to gentamycin, 62% to azithromycin, 55% to ciprofloxacin, 56% to ofloxacin, 43% to Trimethoprim/Sulfamethoxazole, and 41% to clindamycin [36], respectively. These results are opposite to the results of the current study except for chloramphenicol and doxycycline. The difference in the antibiogram was justifiable in terms of the nature of specimens such as poultry, and animal-related infections were the most difficult to treat infections and demand more extended antibiotic therapy and a hospital stay which contributes to the emergence of antibiotic resistance; other factors may include prescription of antibiotics, availability of antibiotics, and biofilm-forming potential of the causative agent.

Biofilm formation involves a complex community of pathogens on the biotic and non-biotic surfaces and are labeled as major contributing factors in the AMR in biofilms [37,38]. A previous study has isolated 305 MRSA and revealed that all strains (100%) were biofilm producers; among them, only 13% were strong biofilm producers [39]. These investigations are in line with the current study outcomes.

A previous study on the biofilm formation potential of MRSA concluded that prevalence of *fib* is 90%, *cna* 93%, and *fnbB* 53% [20], respectively. These results are the same as reported by our study except *cna,* in which we reported 48%. Previous studies showed variable results of biofilm-related genes such as *icaA and icaD* (34%) [17], *clfA* (100%)*, eno* (78%) *clfB* (100%), *fib* (74%), *fnbB* (46%), *fnbA* (56%)*,* and *cna* (54%) [40], respectively. Another study reported the *fnbA* (78%), *fnbB* (81%), *clfA* (59%), and *cna* (73%) in MRSA isolated from pharyngitis patients and have the potential of biofilm production [15]; a recent study reported *clfA* and *clfB* completely dominant followed by f*nbA* (80%), *fnbB* (77%), *sdrC* (68%), *icaA* (63%), *icaD* (58%), *sdrD* (54%), *can* (25%), and *fib* (20%) in biofilm producer MRSA [41], respectively. These dissimilarities in the prevalence of virulence genes might be influenced by genetic factors, epidemiological variations, specimen source, transmission routes, and environmental factors, respectively. 

This study revealed that most of the potent biofilm producer strains of MRSA belonged to SCC*mec* II and *agr*I, moderate biofilm producers belonged to SCC*mec* IV, and *agr* III and weak biofilm producers belonged to SCC*mec* IV and *agr* IV, respectively. A previous study on MRSA isolated from bacteremia patients reported that SCC*mec* IV is predominant in biofilm development [42]. Previous studies involving biofilm formation potential of MRSA and SCC*mec* and *agr* typing demonstrated the same results as indicated in this study with some differences which are justifiable in terms of specimen nature and geographical privileges [14,16,43]. Biofilm formation potential was not significantly attributed to the SCC*mec* and *agr* typing, but both schemes typed all MRSA strains that produced biofilm.

Study Limitations: This current study successfully justified the outcomes and explained the variations among various studies conducted in different parts of the world. However, it has certain limitations. Besides the possibility of bacterial infections, the possibility of other respiratory viral infections, or infections by atypical bacteria, were not investigated, which might be important to rule out the possibility of other lethal bacterial infections. The current study did not report the data on clinical and subclinical conditions of patients because of the restriction in ethical approval from the target institutions. Furthermore, because of the financial limitations, the sequencing analysis of PCR products could not be carried out in order to further identify the isolates as well as to check their phylogenetics. Hence, some factors might be missing that contribute to HAIs. Further large-scale and multi-institutional studies are recommended.

## 5. Conclusions

This genetic characterization of MRSA isolates revealed that most of the potent biofilm producer strains belonged to SCC*mec* II and *agr*I, moderate biofilm producers belonged to SCC*mec* IV, and *agr* III, and weak biofilm producers belonged to SCC*mec* IV and *agr* IV, respectively. Furthermore, the results of the current study significantly contributed to understanding and managing the spectrum of biofilm-producing MRSA-associated infectious diseases probably in hospital settings. The results also provide the potential molecular candidates for biofilm-producing MRSA.

## Figures and Tables

**Figure 1 antibiotics-12-00157-f001:**
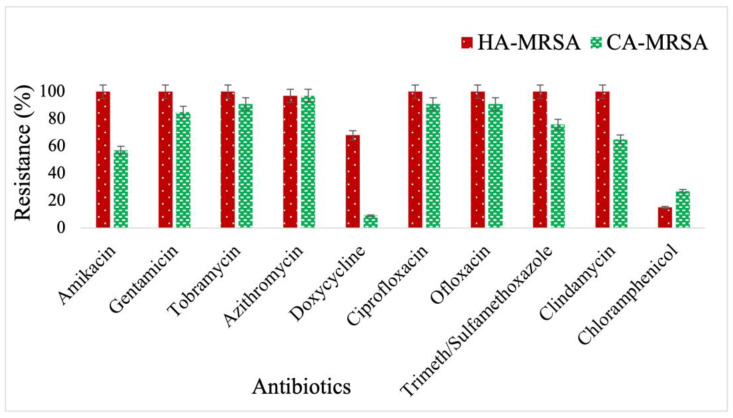
Antibiogram of HA-MRSA and CA-MRSA against recommended antibiotics.

**Figure 2 antibiotics-12-00157-f002:**
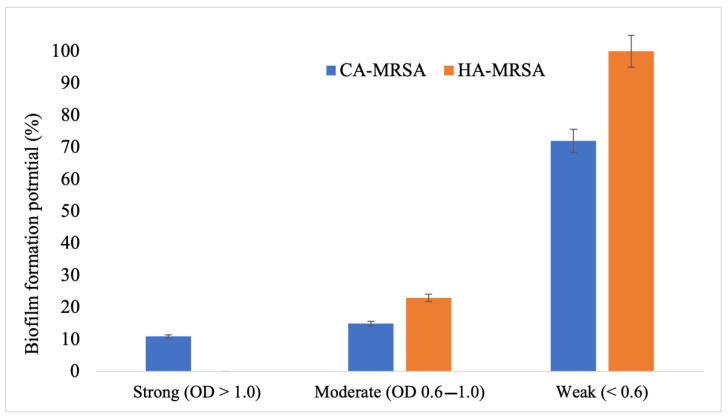
Biofilm production of CA-MRSA and HA-MRSA isolates using the Microtiter Plate method.

**Table 1 antibiotics-12-00157-t001:** Overall prevalence of pathogens isolated from different clinical samples (*n* = 2973).

The Pathogen Identified following CLSI Guidelines		Number (*n*)	% Prevalence
*S. aureus* (*n* = 736)	MRSA	221	7.43
	MSSA	501	16.85
	CONS	14	0.47
*Enterococcus faecalis*		39	1.31
*Klebsiella pneumoniae*		778	26.16
*Acinetobacter baumannii*		336	11.30
*Escherichia coli*		801	26.94
*Pseudomonas aeruginosa*		257	8.64
*Stenotrophomonas maltophilia*		26	0.87

MRSA: Methicillin Resistance *Staphylococcus aureus*. CONS: Coagulase Negative *Staphylococcus aureus*. MSSA: Methicillin Sensitive *Staphylococcus aureus*.

**Table 2 antibiotics-12-00157-t002:** Molecular characterization of biofilm-producing MRSA isolates.

Biofilm	Biofilm Formation Genes	*SCCmec* Typing	*agr* Typing
Strong (OD > 1.0)	*clfA* (21%), *clfB* (24%), *icaAD* (23%), *icaBC* (24%), *eno* (24%), *fnbA* (22%), *fnbB* (20%), *fib* (23%), *sdrC* (25%), *sdrD* (24%), *sdrE* (27%), and *cna* (14%)	*SCCmec* II (25%), *SCCmec* III (15%), *SCCmec* IV (21%), *SCCmec* V (6%)	*agr* I (54%), *agr* II (15%), *agr* III (17%), *agr* IV (14%)
Moderate (OD 0.6–1.0)	*clfA* (27%), *clfB* (28%), *icaAD* (28%), *icaBC* (24%), *eno* (28%), *fnbA* (27%), *fnbB* (23%), *fib* (28%), *sdrC* (24%), *sdrD* (21%), *sdrE* (24%), and *cna* (20%)	*SCCmec* II (15%), *SCCmec* III (13%), *SCCmec* IV (33%), *SCCmec* V (12%)	*agr* I (15%), *agr* II (13%), *agr* III (33%), *agr* IV (13%)
Weak (< 0.6)	*clfA* (52%), *clfB* (49%), *icaAD* (49%), *icaBC* (53%), *eno* (48%), *fnbA* (50%), *fnbB* (57%), *fib* (49%), *sdrC* (51%), *sdrD* (54%), *sdrE* (48%), and *cna* (38%)	*SCCmec* II (14%), *SCCmec* III (21%), *SCCmec* IV (39%), *SCCmec* V (7%)	*agr* I (27%), *agr* II (5%), *agr* III (2%), *agr* IV (67%)

## Data Availability

Data related to the current study can be accessed upon a reasonable request to drsohailmmg@gmail.com.

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
