# Peer review of "Molecular Characterization of Community- and Hospital- Acquired Methicillin-Resistant *Staphylococcus aureus* Isolates during COVID-19 Pandemic"

_antibiotics, 2023, doi:10.3390/antibiotics12010157_

Round 1

Reviewer 1 Report

It is well established that Staphylococcus aureus is a common cause of nosocomial and community-acquired infections. While MRSA is one of the most important pathogens worldwide, some strains are restricted to one geographic area. This makes molecular characterization of great importance for investigating the epidemiology of MRSA both locally and globally. Biofilm formation is considered a virulence property of S. aureus and results in recurrent infections that are difficult to treat and lead to higher treatment costs.Therefore, the importance of this work lies in the search for molecular indicators for the identification of MRSA biofilm-producing through the study of community and hospital acquired MRSA isolates during COVID-19 pandemic.

Suggestions and questions for authors

1) On the page 5, pp174 and pp175,  where it says back should be black.

2) How many Covid-19 patients were found to be positive for MRSA?

3) On the page 5, pp182-183, authors say “No significant difference was found between HA-MRSA and CA-MRSA in terms of biofilm production qualitatively or quantitatively”. However, on the page 7, pp218, thew say “On the other hand, CA-MRSA was a strong biofilm producer as compared to HA-MRSA”. It's confusing.

4) On the page, pp219, the authors say that SCCmec and agr typing did not correlate significantly with the strength of biofilm production, but more than 50% of strong biofilm MRSAs are agr I type. In fact, they make a mention of this in the discussion: "This study revealed that most of the potent biofilm producer strains of MRSA belonged to SCCmec II and argI..." In addition, this study, along with others, reported a relationship between the biofilm formation potential of MRSA and SCCmec IV. The conclusions also highlight the correlation of biofilm production and SCCmec or agr. Could you explain or show your statistical study and the absence of a statistically significant difference between biofilm formation potential and SCCmec or agr typing? Your  conclusions and the claim that no statistically significant difference was found between biofilm formation potential and SCCmec or agr typing is confusing.

5) In the discussion with reference to the paragraph "this difference in prevalence is justifiable based on surgical practices and the tendency to review cases of surgical procedures and health facilities in general", do the authors referring to the difference respect their study? , the prevalence of a study for specific infections (reference 24) is being compared, but this study does not give data about infections correlated with isolated MRSA.

6) The authors should exclude bbp gene prevalence data from the Atshan et al study (reference 15) as this gene is not analyzed in the present study.

Author Response

Reviewer 1

Comments and Suggestions for Authors

It is well established that Staphylococcus aureus is a common cause of nosocomial and community-acquired infections. While MRSA is one of the most important pathogens worldwide, some strains are restricted to one geographic area. This makes molecular characterization of great importance for investigating the epidemiology of MRSA both locally and globally. Biofilm formation is considered a virulence property of S. aureus and results in recurrent infections that are difficult to treat and lead to higher treatment costs. Therefore, the importance of this work lies in the search for molecular indicators for the identification of MRSA biofilm-producing through the study of community and hospital acquired MRSA isolates during COVID-19 pandemic.

Response: Dear reviewer, we would like to thank you for your kind efforts to review our manuscript. We also would like to appreciate that; our manuscript has significantly improved after addressing the comments from you and another reviewer.

Suggestions and questions for authors

1) On the page 5, pp174 and pp175, where it says back should be black.

Response: Line 205: The spelling mistake has been corrected.

2) How many Covid-19 patients were found to be positive for MRSA?

Response: Line 187-188: A new sentence has been added in the revised version of manuscript. “The prevalence of MRSA among COVID-19 patients who were coinfected with different bacterial infections was 17.4% (n = 42).”

3) On the page 5, pp182-183, authors say “No significant difference was found between HA-MRSA and CA-MRSA in terms of biofilm production qualitatively or quantitatively”. However, on the page 7, pp218, thew say “On the other hand, CA-MRSA was a strong biofilm producer as compared to HA-MRSA”. It's confusing.

Response: Dear reviewer, thank you for highlighting the difference. We have removed the sentence at line 182-183 as it was wrongly written and not representing the results.

4) On the page, pp219, the authors say that SCCmec and agr typing did not correlate significantly with the strength of biofilm production, but more than 50% of strong biofilm MRSAs are agr I type. In fact, they make a mention of this in the discussion: "This study revealed that most of the potent biofilm producer strains of MRSA belonged to SCCmec II and argI..." In addition, this study, along with others, reported a relationship between the biofilm formation potential of MRSA and SCCmec IV. The conclusions also highlight the correlation of biofilm production and SCCmec or agr. Could you explain or show your statistical study and the absence of a statistically significant difference between biofilm formation potential and SCCmec or agr typing? Your conclusions and the claim that no statistically significant difference was found between biofilm formation potential and SCCmec or agr typing is confusing.

Response: Dear reviewer, we apologise that we have mentioned here wrongly. Actually, here we wanted to mention about agr II and III. To make it clear for the reader, and to make it as overall agr typing, we have removed this sentence from the discussion section. (Line 219 from the previous version has been removed)

5) In the discussion with reference to the paragraph "this difference in prevalence is justifiable based on surgical practices and the tendency to review cases of surgical procedures and health facilities in general", do the authors referring to the difference respect their study? , the prevalence of a study for specific infections (reference 24) is being compared, but this study does not give data about infections correlated with isolated MRSA.

Response: Line 253-255: The sentence has been revised and the reference study has been changed (Ref 29).

6) The authors should exclude bbp gene prevalence data from the Atshan et al study (reference 15) as this gene is not analyzed in the present study.

Response: Line 287: The prevalence data of bbp gene from the Atshan et al study has been removed from the discussion as this gene was not analyzed in the present study.

Reviewer 2 Report

Dear Authors,

I congratulate all the Authors for their contributions to the writing of the manuscript entitled “Molecular characterization of community and hospital acquired methicillin-resistant Staphylococcus aureus isolates during COVID-19 pandemic”.

I have few comments on the manuscript:

1.  In line 44, the Authors should use the definitive word ‘will’ instead of ‘might’. This will show the importance of the study, by which ‘will’ expresses certainty, while ‘might’ expresses possibility.

2. In third paragraph of the introduction, a more detailed background of the mechanism of biofilm formation will be an added value to the manuscript.

3.   The methods are poorly written. Please elaboratively described methods used in the study (i.e. 2.3. Pathogen isolation and identification). Please also include a table for zone of inhibition used in the study based on CLSI guidelines to guide the readers.

4.    In line 184, n should be written in italics.

5.    Figure 2 – there are no reference to * and ** in the figure 2.

6.    Line 196, SCCmec should be in italics.

7.    Lines 62 and 207, should it be a hyphen between ST-8, ST-30, ST-2490, ST-8, ST-5 and ST-72? Because the Authors wrote ST-239 in line 60. Please be consistent throughout the manuscript.

8.  Please review the entire manuscript and all bacterial species name should be in short form after it was first introduced (i.e. in line 174, 181, etc).

9. Please review the use of spacing, punctuation marks and grammar throughout the manuscript. For example, in lines 32, 123, 127 there should be a space between number and its unit, line 33 en dash () should be used instead of a hyphen (-), line 35 there should be a hyphen between Kirby-Bauer, line 36 biofilm should not be capital, etc.

10. The logical flow of this manuscript is not perfect and awkward. The authors have written several matters randomly. For example, in line 76-77.

11. It is not a good way to introduce a topic by leading with authors' names. i.e. Walls RJ et al. in line 221, Stefánsdóttir A et al. in line 222, Gerardo Alvarez-Uria and Raghuprakash Reddy in line 235, and many other places throughout the manuscript. Concentrate on the point made and then cite the authors at the end of the sentence. Who they are is less important.

12.  Study limitations should be elaborated more.

13.  Please confirm that the study received no external funding and if available, please mention the funding details in line 301.

I consider the manuscript is sufficiently comprehensive, however it needs a thorough check for its grammar/use of punctuations and proper editing/structuring according to these comments. The references are complete and do not need to be included more.

My sincere congratulations to all Authors.

Author Response

Reviewer 2

Comments and Suggestions for Authors

Dear Authors,

I congratulate all the Authors for their contributions to the writing of the manuscript entitled “Molecular characterization of community and hospital acquired methicillin-resistant Staphylococcus aureus isolates during COVID-19 pandemic”.

Response: Dear reviewer, we really appreciate your kind appreciation to our work. We also would like to thank you that your comments have really helped us to improve the quality of manuscript and make it better for the reader.

I have few comments on the manuscript:

  1. In line 44, the Authors should use the definitive word ‘will’ instead of ‘might’. This will show the importance of the study, by which ‘will’ expresses certainty, while ‘might’ expresses possibility.

Response: Line 43: Dear reviewer, thank you for your valuable suggestion. We have replaced “might” with “will” in the revised manuscript.

  1. In third paragraph of the introduction, a more detailed background of the mechanism of biofilm formation will be an added value to the manuscript.

Response: Line 75-92: New information has been added as suggested.

  1.  The methods are poorly written. Please elaboratively described methods used in the study (i.e. 2.3. Pathogen isolation and identification). Please also include a table for zone of inhibition used in the study based on CLSI guidelines to guide the readers.

Response: Line 129-136, Table 1: The methods has been elaborate and a new table has been added to show the zone of inhibitions.

  1. In line 184, nshould be written in italics.

Response: Line 188: “n” has been italicized.

  1. Figure 2 – there are no reference to * and ** in the figure 2.

Response: Dear reviewer, thank you for highlighting the point. These starts were wrongly placed during the layout formatting of manuscript. We have removed these stars from the revised version of manuscript.

  1. Line 196, SCCmecshould be in italics.

Response: Line 225: “mec” has been italicized.

  1. Lines 62 and 207, should it be a hyphen between ST-8, ST-30, ST-2490, ST-8, ST-5 and ST-72? Because the Authors wrote ST-239 in line 60. Please be consistent throughout the manuscript.

Response: Line 65: Dear reviewer, thank you for highlighting the point. We have removed the hyphen at line 65 in order to be consistent throughout the manuscript.

  1. Please review the entire manuscript and all bacterial species name should be in short form after it was first introduced (i.e. in line 174, 181, etc).

Response: Dear reviewer, we have checked the entire manuscript for bacterial names and species names and corrected them wherever needed.

  1. Please review the use of spacing, punctuation marks and grammar throughout the manuscript. For example, in lines 32, 123, 127 there should be a space between number and its unit, line 33 en dash (–) should be used instead of a hyphen (-), line 35 there should be a hyphen between Kirby-Bauer, line 36 biofilm should not be capital, etc.

Response: Dear reviewer, thank you for your valuable suggestions. We have checked the punctuation marks and grammar throughout the manuscript and revised wherever needed.

  1. The logical flow of this manuscript is not perfect and awkward. The authors have written several matters randomly. For example, in line 76-77.

Response: Line 93-98: Dear reviewer, thank you for highlighting the valuable point. We have revised the content in introduction section and have also added some more background.

  1. It is not a good way to introduce a topic by leading with authors' names. i.e. Walls RJ et al. in line 221, Stefánsdóttir A et al. in line 222, Gerardo Alvarez-Uria and Raghuprakash Reddy in line 235, and many other places throughout the manuscript. Concentrate on the point made and then cite the authors at the end of the sentence. Who they are is less important.

Response: Line 249, 250, 264, 269, 286, 290, 293, and 300: Dear reviewer, thank you for your valuable suggestion. We have revised the discussion section accordingly.

  1. Study limitations should be elaborated more.

Response: Line 309-317: Study limitations has been elaborated.

  1. Please confirm that the study received no external funding and if available, please mention the funding details in line 301.

Response: Dear reviewer, the current study has not received any external funding, hence we mentioned it at line 334.

I consider the manuscript is sufficiently comprehensive, however it needs a thorough check for its grammar/use of punctuations and proper editing/structuring according to these comments. The references are complete and do not need to be included more.

My sincere congratulations to all Authors.

Response: Dear reviewer, thank you once again for your kind appreciations, it means a lot to us. We have revised the manuscript as per the comments from your side and other reviewer. Furthermore, we have thoroughly revised the manuscript for English proofreading and grammatical mistakes.

Round 2

Reviewer 2 Report

Again, I congratulate all the Authors for their contributions to the writing of the manuscript.

I believe that the manuscript is now suitable for publication in the Antibiotics journal and can be accepted after minor spell checking/editing.

Author Response

Reviewer 2

Comments and Suggestions for Authors

Again, I congratulate all the Authors for their contributions to the writing of the manuscript. I believe that the manuscript is now suitable for publication in the Antibiotics journal and can be accepted after minor spell checking/editing.

Response: Dear reviewer, thank you again for your valuable comments. We have thoroughly checked the manuscript for English proofreading and grammatical mistakes and revised it wherever needed.